# Small joint arthrodesis technique using a dowel bone graft in a rabbit model

Ki-Tae Na[1][☯], Yoon-Min Lee[2][☯], Jae-Hoon Choi[2], Seok-Whan Song[2]*

**1** Department of Orthopedic Surgery, Suwon Nanoori Hospital, Suwon, Korea, **2** Department of Orthopedic Surgery, Yeouido St. Mary's Hospital, College of Medicine, The Catholic University of Korea, Seoul, Korea

☯ These authors contributed equally to this work.
* sw.song@catholic.ac.kr

**Data Availability Statement:** All relevant data are within the paper and its Supporting Information files.

**Funding:** The authors received no specific funding for this work.

## Abstract

The dowel bone graft fusion technique for the ankle is a well-known and useful method. However, clinical results of dowel bone graft for small joint fusion are unknown. The objective of the present study is to evaluate the effects of dowel bone graft technique for small joint arthrodesis in an *in vivo* arthrodesis of rabbit elbow model compared with the conventional arthrodesis technique (open, joint surface debridement, and internal fixation method). We assigned 28 young adult New Zealand white rabbits to one of two groups: Group 1, the conventional fusion technique group; Group 2, the dowel bone graft fusion technique group. We performed arthrodesis surgery in two different ways for each group. Eight weeks after the operation, specimens were harvested, radiographed, mechanically tested for torque to failure and stiffness, and evaluated for histology. Fusion rates were 77% (10/13) in Group 1 and 93% (13/14) in Group 2 ($p = 0.326$). Torque to failure showed a mean of 0.86 Nm in Group 1 and 0.77 Nm in Group 2 ($p = 0.464$). The mean value of stiffness was 0.11 Nm/deg in Group 1 and 0.11 Nm/deg in Group 2 ($p = 0.832$). In Group 2, histological examination showed residual cartilage absorption and inflammatory response in all cases. In this model, we have been unable to show a difference in either the union rate or strength of fusion between the two methods. However, the dowel bone graft technique is an easy and less invasive method and has some advantages over the conventional method.

## Introduction

As joint replacement techniques and equipment have been developed, the central role of treatment for end-stage arthritis has already moved to joint replacement arthroplasty. However, unlike large joint, small joint arthroplasty has low patient satisfaction, high incidence of complications (deep infection, early aseptic loosening), and requires replacement in time. As a result, arthrodesis still plays a major role in the treatment of small joint end-stage arthritis [1–5].

Small joint arthrodesis can be performed by open or percutaneous methods, internal or external fixation methods, and may or may not include autogenous bone grafting. Most articles presented a relatively high union rate and good clinical results regardless of the arthrodesis technique. However, complications related to arthrodesis are relatively common. [1, 6, 7].

**Competing interests:** The authors have declared that no competing interests exist.

Although finger joint arthrodesis is one of the simplest arthrodesis technique, Stern and Fulton [7] have reported 20 percent major complications (nonunion, malunion, deep infection, and osteomyelitis), and 16 percent minor complications (dorsal skin necrosis, cold intolerance, joint stiffness, paresthesia).

The technique of dowel bone graft has been accepted as an alternative arthrodesis method in the ankle. The concept of dowel bone graft was clinically used for the first time in ankle arthrodesis by Baciu [8] and many modifications have been attempted [9, 10]. However, there is little published literature in English that applied the dowel bone graft technique in a small joint, e.g. a finger joint. We hypothesized that the dowel bone graft technique for small joint arthrodesis could be more effective, simple, and safer. In order to prove the hypothesis, animal experiments were planned.

The purpose of the present study was to evaluate the effects of dowel bone graft technique for small joint arthrodesis in an *in vivo* arthrodesis of rabbit elbow model compared with the conventional arthrodesis technique (open, joint surface debridement, and internal fixation method). We chose the in vivo rabbit elbow arthrodesis model because the size (approximately 10 mm) and the movements (flexion and extension, as a hinge joint) of the rabbit's elbow joint are similar to those of the human phalangeal joint. We therefore determined: (1) the fusion rates of both conventional and dowel bone graft techniques; (2) mechanical properties of fused joints after conventional and dowel bone graft techniques; and (3) the histologic appearance of the fused rabbit elbow, especially focusing on the status of remained cartilage and maturity of the fusion site in Group 2. By confirming the effectiveness of dowel bone graft technique in the rabbit model, we wanted to confirm if the dowel bone graft technique is a good method for small joint fusion.

## Materials and methods

### General study design

This animal care and use protocol was reviewed and approved by the Institutional Animal Care and Use Committee, Yeouido St. Mary's Hospital, The Catholic University of Korea (Approval Number: YEO20162401FA). Twenty-eight skeletally mature New Zealand white rabbits were used. Specific effort was made to minimize the number of animals. All rabbits were older than six months and ranging in size from 2.9 to 3.2 kg. Rabbits were procured from a licensed rabbit farm and were not inbred. All rabbits were kept in a cage under a conventional 12 hour light-dark cycle (7:00 a.m./p.m.), inspected morning and evening, and acclimated to our animal facility for five to seven days before the designated surgical procedure. All animal procedures were carried out in accordance with national ethical guidelines. Humane endpoints for all experiments were defined as 10% acute weight loss or clinical signs consistent with uncontrolled surgical infection. We randomly assigned 28 young adult New Zealand white rabbits into two groups as follows: Group 1, the conventional technique group; Group 2, the dowel bone graft technique group. At 8 weeks after the operation [11] the animals were euthanized by intravenous administration of sodium pentobarbital/isopropyl alcohol solution by the veterinary doctor, and the specimens were harvested. We made every effort to minimize the suffering throughout the experimental process. In this study, radiographic evaluations were carried out to confirm joint fusion, mechanical tests for torque to failure and stiffness, and histologic evaluation (only in Group 2) to assess for remained cartilage status and fusion mass.

### Operative protocol

All rabbits were anesthetized by the veterinary doctor (H. M. Jo) using a combination of Alfaxan 5 mg/kg and Xylazine 5 mg/kg via intramuscular injection, and additional isoflurane

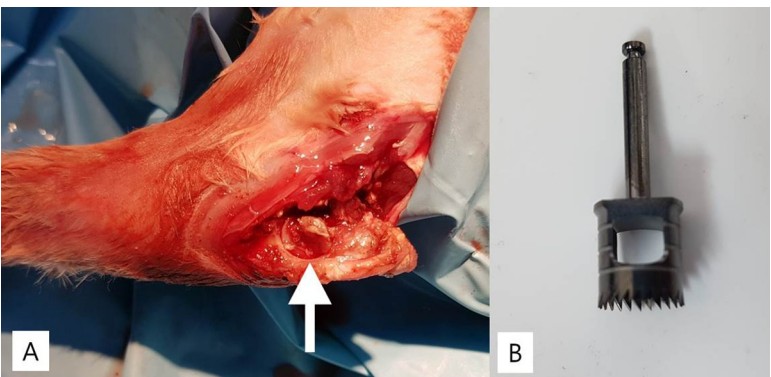

**Fig 1. Intraoperative photographs.** (A) Intraoperative photographs of the *in vivo* elbow arthrodesis procedure in a rabbit. The dowel bone (white arrow) was prepared with trephine and rotated 90 degrees from its natural position. (B) Ø7 mm trephine.

1.5–2% with 100% oxygen was supplied via facemask. The animals were monitored closely during surgery by a veterinary doctor. After induction of anesthesia, prophylactic intramuscular antibiotics (cefazolin, 33 mg/kg) were administrated within 30 minutes of surgery. The current study was initially designed to perform bilateral elbow arthrodesis. However, rabbits did not tolerate bilateral procedures in the pilot test. Therefore, the rabbit's left forelimb was chosen to eliminate technical differences due to right and left side procedures.

The operative limb was shaved and sterile skin preparation was done with povidone and 90% alcohol. Rabbits were placed in the lateral decubitus position and a lateral approach was used to expose the elbow joint. Before skin incision, subcutaneous lidocaine was used for pain control. An oblique skin incision was made over the elbow, and soft tissue dissection was done. Triceps brachii muscle and surrounding musculatures were detached from the olecranon and forelimb bone (Fig 1).

## Conventional technique

Cartilages on the articular surfaces of the ulna and humerus were removed meticulously using #11 blades and curettes. Subchondral bones were also removed until the cancellous bone appeared. Both the proximal ulna and distal humerus were trimmed to congruent contact surfaces. The above procedures were carefully done to prevent proximal ulnar fracture. While maintaining elbow alignment in 90 degree flexion, two or three Kirschner's wires were used to stabilize the joint.

## Dowel bone graft

After exposing the elbow joint, we harvested a round dowel bone graft (Fig 1) from an elbow joint with a Ø 7 mm trephine drill (Cowellmedi, Busan, Korea) (Fig 1). Drilling was done with care, not to shatter the proximal ulna. Continuous irrigation was performed during drilling by using a syringe to prevent thermal injury. The harvested dowel graft was rotated 90 degrees counterclockwise to reposition the original joint surface of the joint in the vertical plane, instead of in the bone-cartilage-bone horizontal joint plane configuration (Fig 2). Then, two or three Kirschner's wire fixations were performed at 90 degree elbow flexion (Fig 2). Rigid fixation was obtained in all cases.

All joint arthrodesis was performed at 90 degrees of joint flexion to approximate the normal functioning position of the rabbit elbow. Operative wounds were closed with a layer-by-layer

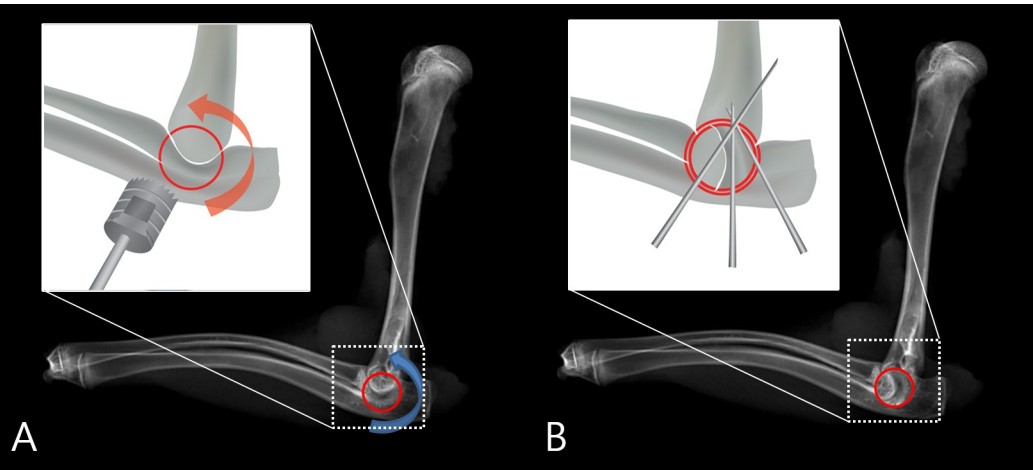

**Fig 2. Schematic diagrams of dowel bone graft technique.** (A) A round dowel bone graft was harvested from an elbow joint (red ring) and was rotated 90 degrees counterclockwise (curved arrow). (B) The rotated joint was placed in the vertical plane. Three Kirschner's wires were used to fix joint.

technique. The forelimb was immobilized using a padded forelimb spica cast. After surgery, the rabbits were given cefazolin 33 mg/kg every 12 hours for five days. Subcutaneous Ketorolac was injected for pain control for two days. Rabbits were allowed to walk as tolerated on their casts.

## Manual test and radiographic evaluation

Animals were euthanized at 8 weeks postoperatively. The forelimbs were disarticulated from the shoulder joint. The harvested forelimbs were stripped of all soft tissue from the humerus to the forearm, K-wires were removed last not to disrupt the fusion site during stripping. Limbs were placed in saline soaked gauze until the test began. A manual test was performed to check the gross motion across the fusion site. Gross motion across the fusion site was defined as any slight motion during flexion-extension movement of the elbow joint with slight motions of the upper and forearms. After that, lateral X-rays were taken to assess radiographic union, followed by mechanical testing. Successful fusion was defined as no gross motion across the fusion site in the manual test and a union across both cortices on lateral X-rays of the elbow.

## Mechanical testing

All mechanical testing was done within three hours of harvest. Rabbit forelimbs were refrigerated while awaiting testing and acclimated to room temperature in the testing facility. Each fused forelimb was tested for torque to failure and stiffness using an Instron® testing apparatus (Instron, Norwood, MA, USA). A custom jig system was used to mount the rabbit forelimb on an Instron® testing apparatus (Fig 3). The jig consisted of a steel cylinder with multiple fixation points and a steel body to mount the testing apparatus.

Forelimbs were securely fixed to the testing apparatus and were tested in extension using the precision motor testing device. With the humerus oriented vertically, the ulna was loaded as a cantilever using an unconstrained load. Extension load was accomplished at a constant rate of 0.5 mm/sec. The lever arm was measured from the center of the jig cylinder to the testing arm of the motor testing device (35 mm). Torque to failure and stiffness of the fused joints were calculated from the testing device and recorded. Torque to failure was defined as the point of inflection on the recorded test. At this inflection, torque to failure typically decreased

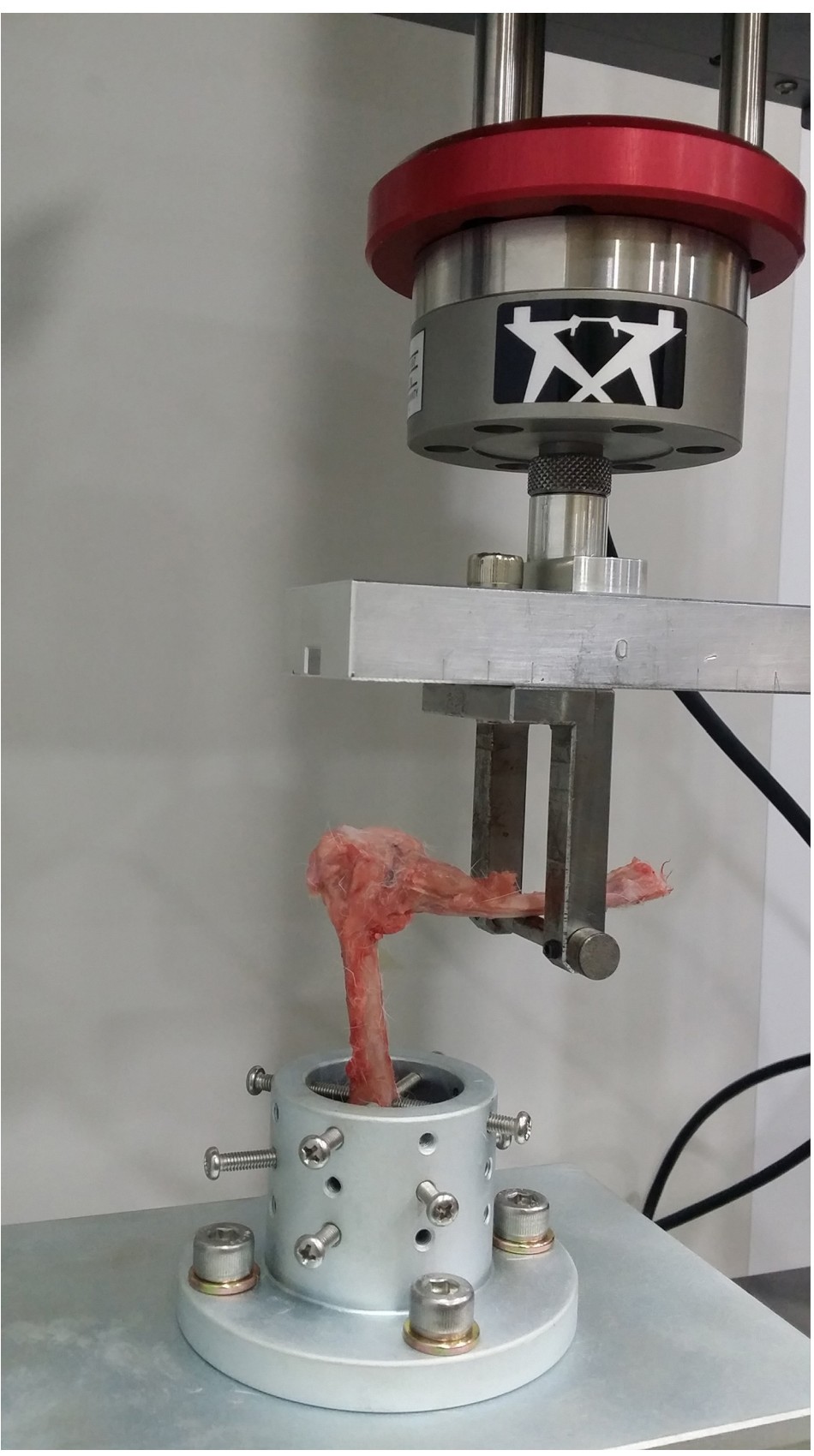

**Fig 3. The custom testing jig and Instron® testing apparatus.** The humerus was fixed in an iron mounting fixture with multiple screws. The horizontal bar was attached to a calibrated motor that generated extension force on the ulna.

as the displacement progressed. After mechanical testing, rabbit forelimbs were placed in 10% neutral formalin for histologic evaluation.

## Histologic evaluation

After demineralization, the specimens of Group 2 were placed into paraffin wax and sectioned through the sagittal plane of the joint. Hematoxylin and Eosin (H & E) staining were performed for every specimen. Slides were examined for fusion status and remained cartilage changes. The S-100 protein immunostaining was performed to assess the status of chondrocytes. Articular cartilage, which is contained in the dowel bone graft, was examined. All histologic evaluations were done by an experienced pathologist (Dr. T. J. K), blinded to our hypothesis.

## Statistical analysis

Statistical analyses were performed using IBM SPSS Statistics for Windows, Version 21.0 (IBM Corp., Armonk, NY, USA). The descriptive evaluation was presented on the means with standard deviation, frequencies, or percentages. Fusion rate between groups was compared using Fisher's exact test. Torque to failure and stiffness between groups were compared using two-sided Student's t-test. The level of significance was set at $P < .05$.

## Results

One rabbit in Group 1 developed an infection at the arthrodesis site and was excluded from the study. The remaining 27 rabbits were included and manual, radiographic, biomechanical, and histologic examinations were performed.

### Fusion

Overall fusion rates were 77% (10/13) in Group 1 and 93% (13/14) in Group 2 (Fig 4). In Group 1, one of the three failed joints appeared to be fused in the X-ray, but there was gross motion in a manual test. The other two unfused cases in Group 1 failed to show evidence of fusion in a radiographic study and manual test. One unfused case in Group 2 showed gross motion at the elbow and no evidence of fusion in a radiographic evaluation. There was no statistical difference in fusion rate between techniques ($p = 0.326$). The overall fusion rate in the current study was 85%. It was similar to the previously reported fusion rate of a rabbit ulnohumeral arthrodesis model [11].

### Mechanical test

All specimens were tested for torque to failure and stiffness regardless of whether they were fused or not. Fig 5 shows mean torque to failure and stiffness of fused elbows. Mean torque to failure of fused elbows between Group 1 and Group 2 was not significantly different (Group 1; mean: 0.86 Nm, range: 0.55 to 1.35 Nm) (Group 2; mean: 0.77 Nm, range: 0.47 to 1.1 Nm) ($p = 0.464$).

Mean stiffness was also not significantly different between the two fused groups (Group 1; mean: 0.11 Nm/deg, range: 0.05 to 0.20 Nm/deg) (Group 2; mean: 0.11 Nm/deg, range 0.06 to 0.18 Nm/deg) ($p = 0.832$).

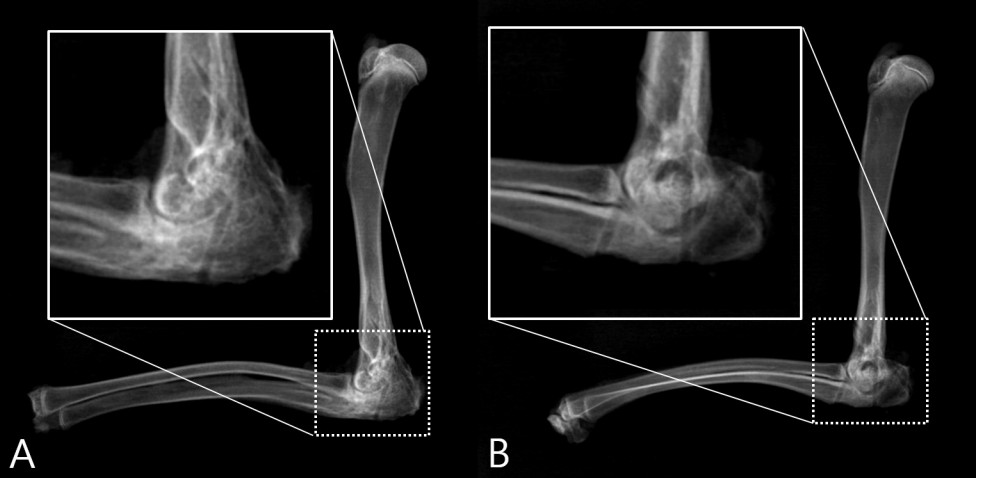

**Fig 4. Simple lateral radiographs.** (A) Lateral radiograph demonstrating solid fusion in Group 1. Well matured fusion mass is seen in a posterior ulnohumeral joint. (B) Lateral radiograph demonstrating solid fusion in Group 2. Fusion mass is seen in both the anterior and posterior ulnohumeral joint.

Regardless of group, the mean torque to failure of four unfused joints was lower than fused joints (fused joint; 0.81 Nm, unfused joint; 0.48 Nm). The mean stiffness of four unfused joints was lower than fused joints (fused joint; 0.11 Nm/deg, unfused joint; 0.03 Nm/deg). The stiffness of unfused joints was always less than 0.05 Nm/deg. On the contrary, that of fused joints was almost always more than 0.05 Nm/deg. Due to the small number of unfused joints, we could not statistically compare the mechanical properties between fused and unfused joints.

## Histologic analysis

Histologic examination of the dowel bone graft technique group showed evidence of absorption of the remaining cartilage (Fig 6). Inflammation was found around remaining cartilage in a majority of cases. The chondrocytes were pale and cartilages were absorbed by macrophage-like cells. In S-100 protein immunostaining analysis, chondrocytes in remained cartilage were stained weakly for S-100 protein. The chondrocyte at the central portion of remained cartilage was more weakly stained than the chondrocyte at the peripheral portion (Fig 7).

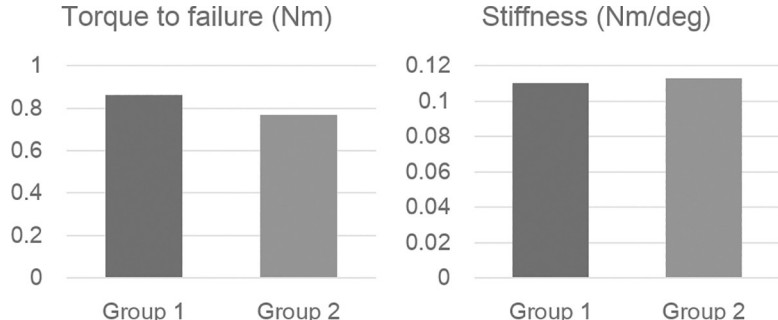

**Fig 5. Torque to failure and stiffness.** Mean values of torque to failure and stiffness of Group 1 and Group 2 for fused specimens.

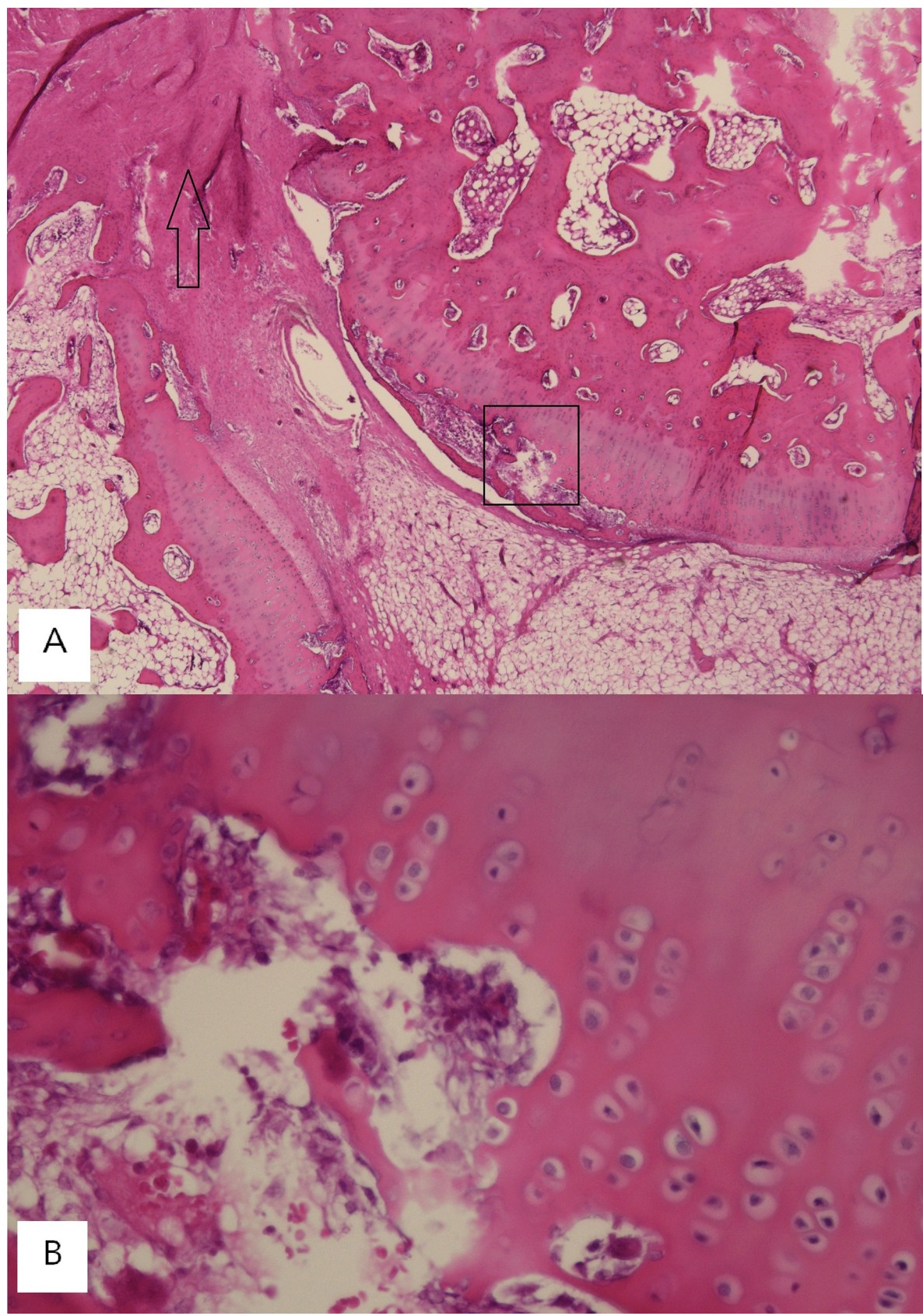

**Fig 6. Histology (H&E).** (A) Low-power magnification (X40). H & E staining of dowel bone graft specimen shows remained cartilage and cartilage absorption (open box) and mature fusion mass around the remained cartilage (open arrow). (B) High-power magnification (X200, H & E staining) of open box in Fig 6A shows cartilage absorption by multinucleated macrophage-like cells.

## Discussion

Arthrodesis is still the most reliable procedure for small joint end-stage arthritis. Regardless of the arthrodesis method, small joint fusion has shown good clinical results. However, it is technically not easy to fuse small joints, and the complication rate is relatively high [6, 7]. In the current study, joint fusion with dowel bone graft technique was easy to implement and provided relatively consistent results without complications in comparison with the conventional technique.

The dowel bone graft technique was used to obtain fusion of ankle joints by Baciu in 1986 [8]. After his first attempt to fuse an ankle joint by dowel bone graft, several authors reported clinical results of dowel ankle arthrodesis [8, 9, 12, 13]. Some physicians tried to introduce dowel bone graft technique to other joint fusion, including tarsometatarsal joint, and sacroiliac joint [10, 14]. We have described a small joint arthrodesis technique using dowel bone graft, which can be especially useful in finger joints. In spite of no statistical significance, our study results showed a higher fusion rate in the dowel bone graft group than the conventional technique group. Mechanical properties were similar in the dowel bone graft and conventional technique groups.

Even if there is no statistical difference in fusion rate and mechanical properties between the two groups, dowel bone graft technique has some advantages. First, the dowel bone graft technique can significantly reduce the need for additional autogenous or allogenic bone grafts, because it utilizes original bone stock. Second, dowel bone graft reduces surgical injury to surrounding soft tissues, because it needs only limited incision and dissection. Third, the dowel bone graft technique also makes it possible to maintain the length of the joint because it does not remove or discard the bone stock. In the conventional technique group, articular cartilages and subchondral bones are removed to make healthy cancellous bone contact with each other and allow fusion to proceed. Excessive bone removal can result in loss of healthy cancellous bone, which can shorten the fused joint.

Dowel bone graft technique requires a specially shaped trephine. In our current study, we used a trephine burr commonly used in dentistry to make dowel bone grafts (Fig 1). When we were starting to make a dowel bone graft using a trephine burr without a guide, the trephine burr moved and made it difficult to drill in the proper position. To overcome this problem, we started drilling from a very low speed and overcame the technical difficulty. There is another technique to treat this problem. If we had used a cannulated trephine burr with a Kirschner's wire as a guide, we could have made the dowel bone graft more easily.

In the current study, remained chondrocytes in dowel bone graft were weakly stained by immunohistochemical staining for the S-100 protein. The function of S-100 protein in chondrocytes still remains unclear. Earlier studies regarded S-100 protein as a marker for chondrocytic phenotype or chondrogenic origin [15–17]. However, recent studies showed that the S-100 protein may be involved in the cartilage repair process [18, 19]. The S-100 protein was stained strongly in degenerating and hypertrophic zones and stained weakly in the proliferating zone of cartilage [20]. In osteoarthritic cartilage, the S-100 protein staining of cartilage has been seen in degenerative regions and osteophytic regions [21]. In our study, we consider the S-100 protein to be stained weakly because the remaining chondrocytes are no longer supplied with oxygen and nutrients from the joint fluid. However, further study will be needed to confirm the reason for weak staining of S-100 protein in remained chondrocytes.

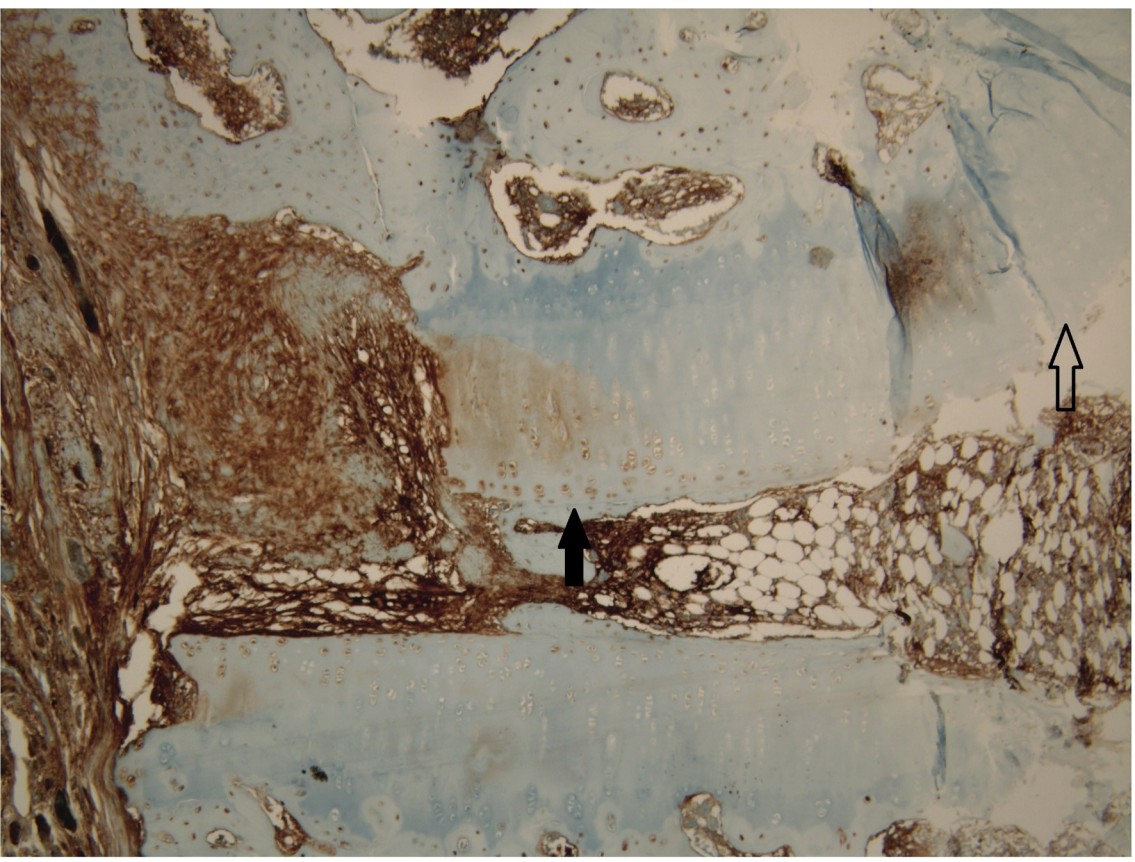

**Fig 7. Histology (S-100 protein).** High-power magnification (X100). The chondrocytes in remained cartilage were weakly stained for S-100 protein. On S-100 protein immunohistochemical staining, the chondrocytes at the central portion of remained cartilage (empty arrow) was more weakly stained than the chondrocytes at the peripheral portion (black arrow).

Although dowel bone graft technique in our study showed satisfactory fusion rate and strong mechanical properties in rabbit elbow arthrodesis model, this study had some limitations. First, our study was performed in a normal joint. The arthritic joint is often deformed and bone quality in some disease would be much worse than a normal joint. In this deformed joint, it would be more difficult to make a dowel bone graft and fix it firmly. However, these difficulties are also encountered in other techniques in treating arthritis with joint deformation. Nevertheless, as long as we have sound bone contact through the circle of dowel bone graft, we can achieve sound fusion.

Second, to identify definitive histologic changes in remained cartilages, the sample size of the current study was too small to make a comparative analysis, and the follow-up period was too short. Larger and long-term studies will be needed in the future.

Third, there are two sites which should be fused in arthrodesis using dowel bone graft. In the conventional technique group, fusion only occurs at the site where the joint was previously present. However, the dowel bone graft technique creates two cut planes (the proximal and distal part of the dowel bone graft), and fusion should occur in both planes. However, if a rigid fixation can be obtained, as with double osteotomy using Ilizarov technique or comminuted fracture with multiple fracture fragments, the fusion can be achieved in both sites simultaneously. The current study also has shown that there is no difference in fusion rate and fusion strength between the two different technique groups using the same fixation technique. The

fact that fusion should occur in two places is not a serious problem if adequate fixation is applied.

Fourth, the lack of histologic analysis prior to mechanical testing was another limitation. It was difficult to analyze the fusion status histologically because the histological examination was performed after the mechanical test. Furthermore, because of the short duration of the experiment, it was unable to observe the complete absorption of cartilage at the arthrodesis.

## Conclusions

In this paper, we have been unable to show a difference in either union rate or strength of fusion between the two methods. However, dowel bone graft technique proved to be an easier and less invasive method and had some advantages over the conventional method. Dowel bone graft technique could be a useful method for arthrodesis of small joints.

## Supporting information

**S1 File. Raw data.** Raw data for fusion and mechanical tests.
(XLSX)

**S2 File. NC3Rs ARRIVE guideline checklist.** A compilation of requirements in reporting in vivo experiments in animals.
(PDF)

## Acknowledgments

We would like to express our sincere gratitude to Jeil Medical Corporation for helping in performing mechanical tests and Kunwoo Company for manufacturing the custom jig system.

We also thank Dr. Tae-jung Kim for his helpful histologic evaluation, and veterinarian Hyun-Moo Jo who helped with the animal experiment overall.

## Author Contributions

**Conceptualization:** Ki-Tae Na, Yoon-Min Lee, Jae-Hoon Choi, Seok-Whan Song.

**Data curation:** Ki-Tae Na, Yoon-Min Lee, Seok-Whan Song.

**Formal analysis:** Ki-Tae Na, Yoon-Min Lee, Seok-Whan Song.

**Funding acquisition:** Ki-Tae Na.

**Investigation:** Ki-Tae Na, Yoon-Min Lee, Jae-Hoon Choi, Seok-Whan Song.

**Methodology:** Ki-Tae Na, Yoon-Min Lee, Seok-Whan Song.

**Project administration:** Ki-Tae Na, Yoon-Min Lee, Seok-Whan Song.

**Resources:** Ki-Tae Na, Yoon-Min Lee, Seok-Whan Song.

**Supervision:** Yoon-Min Lee, Seok-Whan Song.

**Validation:** Ki-Tae Na, Yoon-Min Lee, Jae-Hoon Choi, Seok-Whan Song.

**Visualization:** Ki-Tae Na, Yoon-Min Lee, Jae-Hoon Choi, Seok-Whan Song.

**Writing – original draft:** Ki-Tae Na, Yoon-Min Lee, Jae-Hoon Choi, Seok-Whan Song.

**Writing – review & editing:** Ki-Tae Na, Yoon-Min Lee, Seok-Whan Song.

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
