## [Decision Letter · Decision Letter 0]

10 Oct 2019

PONE-D-19-16141

Small joint arthrodesis technique using a dowel bone graft in a rabbit model

PLOS ONE

Dear Dr. Song,

Thank you for submitting your manuscript to PLOS ONE. After careful consideration, we feel that it has merit but does not fully meet PLOS ONE’s publication criteria as it currently stands. Therefore, we invite you to submit a revised version of the manuscript that addresses the points raised during the review process.

Your study is interesting and may be improved according the reviewers comments. Please get help from a native speaker !

We would appreciate receiving your revised manuscript by Nov 24 2019 11:59PM. To enhance the reproducibility of your results, we recommend that if applicable you deposit your laboratory protocols in protocols.io, where a protocol can be assigned its own identifier (DOI) such that it can be cited independently in the future. For instructions see: http://journals.plos.org/plosone/s/submission-guidelines#loc-laboratory-protocols

We look forward to receiving your revised manuscript.

Kind regards,

Hans-Peter Simmen, M.D., Professor of Surgery

Academic Editor

PLOS ONE

Journal Requirements:

1. Please complete and submit a copy of the ARRIVE Guidelines checklist, a document that aims to improve experimental reporting and reproducibility of animal studies for purposes of post-publication data analysis and reproducibility: https://www.nc3rs.org.uk/arrive-guidelines. Please include your completed checklist as a Supporting Information file. Note that if your paper is accepted for publication, this checklist will be published as part of your article.

Specifically, please ensure that you revise your methods section to include the method of euthanasia and any anaesthesia used, as well as how frequently the condition of the animals was monitored.

3. Please upload a copy of Figure 8, to which you refer in your text on page 14. If the figure is no longer to be included as part of the submission please remove all reference to it within the text.

Additional Editor Comments (if provided):

Both reviewers confirm that your study is interesting and worth to be published. However, in its present form the paper is not ready for final acceptance.

Please, give answers to the reviewers comments. I am convinced that your manuscript will be improved.

You should seek advice from a native speaker since the manuscript is not written in standard english, e.g. lines 47-50, 54, 56, 59, 62, 71-73, 78 (older than, instead of grater than), 81 movement is or movements are and so on.

We are looking forward to receive your revised manuscript.

Reviewers' comments:

Reviewer's Responses to Questions

**Comments to the Author**

1. Is the manuscript technically sound, and do the data support the conclusions?

Reviewer #1: Yes

Reviewer #2: Yes

2. Has the statistical analysis been performed appropriately and rigorously? 

Reviewer #1: Yes

Reviewer #2: I Don't Know

3. Have the authors made all data underlying the findings in their manuscript fully available?

Reviewer #1: Yes

Reviewer #2: Yes

4. Is the manuscript presented in an intelligible fashion and written in standard English?

Reviewer #1: No

Reviewer #2: Yes

5. Review Comments to the Author

Reviewer #1: The authors should clarify how the gross motion across fusion site was tested during the manual test. Has there been a reliable grading of stability? What does "no gross motion" during manual testing mean? Are there any objective measurements so that the authors could clearly define successful fusion?

The authors should clarify the definition of failure. How was the point of inflection exactly defined (load decrease of x%?).

Furthermore, the manuscript should be reworked in order to correct grammatical errors.

Reviewer #2: Dear Colleagues, thank You for the opportunity to review this manuscript.

Abstract: Statistial methods are lacking. Significance levels are lacking. Already in the Abstract the significance has to be stated.

Introduction: What is Your chosen conventional technique? Elaborate linguistically more your scientific aim.

Methods: Here is the conventional technique, but has to be mentioned in three word in the introduction. But OK.

Results: You do not have significant differences between the methods. Histology seems also not to speak a clear language.

Discussion: Too long for what it is but correct.

General: I You would have chosen a much bigger sample, probably than the results would turn significant. More interesting would be which method has to be chosen in chronically ill paients, e.g. Diabetes, HIV, morbid obesity etc.

6. PLOS authors have the option to publish the peer review history of their article (what does this mean?). If published, this will include your full peer review and any attached files.

Reviewer #1: No

Reviewer #2: Yes: Ladislav Mica

---

## [Author Response · Author response to Decision Letter 0]

26 Nov 2019

PONE-D-19-16141

Small joint arthrodesis technique using a dowel bone graft in a rabbit model

PLOS ONE

First of all, thank you very much for your comprehensive review and encouraging comments. We revised the manuscript in accordance with your suggestions. The responses to your comments are appended point-by-point and all the changes are also Yellow Text Highlight Color in the ‘Revised Manuscript with Track Changes’ version.

1. To enhance the reproducibility of your results, we recommend that if applicable you deposit your laboratory protocols in protocols.io, where a protocol can be assigned its own identifier (DOI) such that it can be cited independently in the future.

Unfortunately our laboratory protocol was written in Korean, and we think “Materials and Methods” can be used as a protocol to repeat this study again. If, you would like the protocol of this animal study in Korean and ask to deposit in protocols.io, we will proceed as asked.

2. Please complete and submit a copy of the ARRIVE Guidelines checklist, a document that aims to improve experimental reporting and reproducibility of animal studies for purposes of post-publication data analysis and reproducibility: https://www.nc3rs.org.uk/arrive-guidelines. Please include your completed checklist as a Supporting Information file. Note that if your paper is accepted for publication, this checklist will be published as part of your article.

Specifically, please ensure that you revise your methods section to include the method of euthanasia and any anaesthesia used, as well as how frequently the condition of the animals was monitored.

We are submitting a completed checklist of ARRIVE Guidelines as a Supporting Information file (S2 File). We added the method of euthanasia, and how frequently the condition of the animals was monitored.

All rabbits were kept in a cage under a conventional 12 hour light-dark cycle (7:00 a.m./p.m.), inspected at morning and evening, and acclimated to our animal facility for five to seven days before the designated surgical procedure.

Page 5 Line 73-77

At 8 weeks after the operation [11] the animals were euthanized by intravenous administration of sodium pentobarbital/isopropyl alcohol solution by the veterinary doctor, and the specimens were harvested. We made every effort to minimize the suffering throughout the experimental process.

Page 5 Line 80-84

 We have no restrictions on sharing a de-identified data set. So, we have uploaded an anonymized data set at Support Information file (S1 File)

4. Please upload a copy of Figure 8, to which you refer in your text on page 14. If the figure is no longer to be included as part of the submission please remove all reference to it within the text.

We are sorry for the confusion. Figure 8 is the old name of Figure 6. So, we have changed the text.

(B) High-power magnification (X200, H & E staining) of open box in Figure 8A shows cartilage absorption by multinucleated macrophage-like cells.

(B) High-power magnification (X200, H & E staining) of open box in Figure 6A shows cartilage absorption by multinucleated macrophage-like cells.

Page 13 Line 231-233

Additional Editor Comments (if provided):

Both reviewers confirm that your study is interesting and worth to be published. However, in its present form the paper is not ready for final acceptance.

Please, give answers to the reviewers comments. I am convinced that your manuscript will be improved.

You should seek advice from a native speaker since the manuscript is not written in standard English, e.g. lines 47-50, 54, 56, 59, 62, 71-73, 78 (older than, instead of greater than), 81 movement is or movements are and so on.

Thank you for your kind review of our manuscript. We have changed the English expressions in this paper to standard English with the advice of a native speaker.

However, unlike large joint, arthroplasty of the small joint has shown low patient satisfaction, high complication rate (deep infection, early aseptic loosening), and lack of longevity. As the results, arthrodesis still plays a major role in the treatment of small joint end-stage arthritis.

However, unlike large joint, small joint arthroplasty has low patient satisfaction, high incidence of complications (deep infection, early aseptic loosening), and requires replacement in time. As a result, arthrodesis still plays a major role in the treatment of small joint end-stage arthritis [1-5]. 

Page 3, Line 36-39

However, complication rates associated with arthrodesis still are not uncommon.

However, complications related to arthrodesis are relatively common.

Page 3, Line 43

Even in the case of arthrodesis of finger joints, which is the simplest arthrodesis technique, Stern and Fulton [7] has reported 20 percent of major complications (nonunion, malunion, deep infection, and osteomyelitis), and 16 percent of minor complications (dorsal skin necrosis, cold intolerance, joint stiffness, paresthesia).

Although finger joint arthrodesis is one of the simplest arthrodesis technique, Stern and Fulton [7] have reported 20 percent major complications (nonunion, malunion, deep infection, and osteomyelitis), and 16 percent minor complications (dorsal skin necrosis, cold intolerance, joint stiffness, paresthesia).

Page 3, Line 44-47

Dowel bone graft technique was accepted as an alternative arthrodesis method in an ankle.

The technique of dowel bone graft has been accepted as an alternative arthrodesis method in the ankle.

Page 3, Line 48-49

However, there is no varied English published literature that applied dowel bone graft technique in the small joint, e.g. finger joint.

However, there is little published literature in English that applied the dowel bone graft technique in a small joint, e.g. a finger joint. 

Page 3, Line 50-52

By confirming the effectiveness of dowel bone graft technique in the rabbit model, we wanted to know whether the dowel bone graft technique is a good method for small joint fusion or not.

By confirming the effectiveness of dowel bone graft technique in the rabbit model, we wanted to confirm if the dowel bone graft technique is a good method for small joint fusion.

Page 4, Line 64-66

All rabbit was greater than six months old and ranging in size from 2.9 to 3.2 kg.

All rabbits were older than six months and ranging in size from 2.9 to 3.2 kg.

Page 5, Line 72

In vivo rabbit, elbow arthrodesis model was chosen because the size (approximately 10 mm) and the movement (flexion and extension, as a hinge joint) of the rabbit’s elbow joint are similar to those of human phalangeal joint.

We chose the in vivo rabbit elbow arthrodesis model because the size (approximately 10 mm) and the movements (flexion and extension, as a hinge joint) of the rabbit’s elbow joint are similar to those of human phalangeal joint.

Page 4, Line 58-60

Reviewers' comments:

5. Review Comments to the Author

Reviewer #1: The authors should clarify how the gross motion across fusion site was tested during the manual test. 

First of all, thank you very much for your comprehensive review and encouraging comments. We have clarified how we checked the gross motion in the manuscript. 

A manual test was performed to check the gross motion across the fusion site. After then, lateral X-rays were taken to assess radiographic union, followed by mechanical testing.

A manual test was performed to check the gross motion across the fusion site. Gross motion across the fusion site was defined as any slight motion during flexion-extension movement of the elbow joint with slight motions of the upper and forearms. After that, lateral X-rays were taken to assess radiographic union, followed by mechanical testing. 

Page 8, Line 140-144

Has there been a reliable grading of stability? What does "no gross motion" during manual testing mean?

There is no reliable grading system for stability of fused small joints nor an accurate definition for “gross motion”. We determined “gross motion” if there was any slight visual movement. 

Are there any objective measurements so that the authors could clearly define successful fusion?

We defined successful fusion as no gross motion across the fusion site in the manual test and a union across both cortices on lateral X-rays of the elbow. If either condition was not met, it was not considered a successful fusion.

The authors should clarify the definition of failure. How was the point of inflection exactly defined (load decrease of x%?).

Failure was defined as the point of inflection on the test, where instead of load increases with displacement, load typically decreased.

Torque to failure was defined as the point of inflection on the recorded test. At this inflection, torque to failure typically decreased as the displacement progressed.

Page 9, Line 164-166

Furthermore, the manuscript should be reworked in order to correct grammatical errors.

We have changed the English expressions in this paper to standard English with the advice of a native speaker.

Reviewer #2: Dear Colleagues, thank You for the opportunity to review this manuscript.

 First of all, thank you very much for your comprehensive review and encouraging comments. We have substantially revised the manuscript in accordance with your suggestions. 

Abstract: Statistial methods are lacking. Significance levels are lacking. Already in the Abstract the significance has to be stated.

Statistical methods are described as the last part of Materials and methods. We added p values for all results, so readers can know the significance level of each results. 

Fusion rates were 77% (10/13) in Group 1 and 93% (13/14) in Group 2 (p=0.326). Torque to failure showed a mean of 0.86 Nm in Group 1 and 0.77 Nm in Group 2 (p=0.464). The mean value of stiffness was 0.11 Nm/deg in Group 1 and 0.11 Nm/deg in Group 2 (p=0.832).

Page 2, Line 24-27

Introduction: What is Your chosen conventional technique? Elaborate linguistically more your scientific aim.

The purpose of the present study was to evaluate the effects of dowel bone graft technique for small joint arthrodesis in an in vivo arthrodesis rabbit model.

The purpose of the present study was to evaluate the effects of dowel bone graft technique for small joint arthrodesis in an in vivo arthrodesis of rabbit elbow model compared with the conventional arthrodesis technique (open, joint surface debridement, and internal fixation method).

Page 4, Line 55-58

Methods: Here is the conventional technique, but has to be mentioned in three word in the introduction. But OK.

Thank you for your comment. We have mentioned the three techniques in the introduction.

The purpose of the present study was to evaluate the effects of dowel bone graft technique for small joint arthrodesis in an in vivo arthrodesis rabbit model compared with the conventional arthrodesis technique (open, joint surface debridement, and internal fixation method).

Page 4, Line 55-58

Results: You do not have significant differences between the methods. Histology seems also not to speak a clear language.

Unfortunately, we could not show significant differences between two methods. However, we think we proved the dowel bone graft technique is not inferior to the conventional method in terms of mechanical aspects. Furthermore, dowel bone graft technique has several advantages over the conventional method in clinical aspects. 

Discussion: Too long for what it is but correct.

Thank you for your kind comment. 

General: I You would have chosen a much bigger sample, probably than the results would turn significant. More interesting would be which method has to be chosen in chronically ill patients, e.g. Diabetes, HIV, morbid obesity etc.

Thank you for your recommendation. We strongly hope to have this opportunity in the future. 

6. PLOS authors have the option to publish the peer review history of their article (what does this mean?). If published, this will include your full peer review and any attached files.

Thank you for your explanation. We would like to publish the peer review history of this article, so we say “Yes”.

While revising your submission, please upload your figure files to the Preflight Analysis and Conversion Engine (PACE) digital diagnostic tool, https://pacev2.apexcovantage.com/.

We have uploaded our figure files to the PACE digital diagnostic tool.

---

## [Editor Report · Decision Letter 1]

11 Dec 2019

Small joint arthrodesis technique using a dowel bone graft in a rabbit model

PONE-D-19-16141R1

Dear Dr. Song,

We are pleased to inform you that your manuscript has been judged scientifically suitable for publication and will be formally accepted for publication once it complies with all outstanding technical requirements.

With kind regards,

Hans-Peter Simmen, M.D., Professor of Surgery

Academic Editor

PLOS ONE
---

## [Editor Report · Acceptance letter]

16 Dec 2019

PONE-D-19-16141R1 

Small joint arthrodesis technique using a dowel bone graft in a rabbit model 

Dear Dr. Song:

I am pleased to inform you that your manuscript has been deemed suitable for publication in PLOS ONE. Congratulations! Your manuscript is now with our production department. 

With kind regards,

on behalf of

Dr. Hans-Peter Simmen 

Academic Editor

PLOS ONE